# Nanomaterials Used in the Preparation of Personal Protective Equipment (PPE) in the Fight against SARS-CoV-2

**Pierantonio De Luca** [1,*], **Janos B.Nagy** [1] **and Anastasia Macario** [2]

1   Department of Mechanical and Energy Engineering, University of Calabria, 87036 Arcavacata, Italy; janos.bnagy1@gmail.com
2   Department of Environmental Engineering, University of Calabria, 87036 Arcavacata, Italy; anastasia.macario@unical.it
*   Correspondence: pierantonio.deluca@unical.it

**Abstract:** Following the well-known pandemic, declared on 30 January 2020 by the World Health Organization, the request for new global strategies for the prevention and mitigation of the spread of the infection has come to the attention of the scientific community. Nanotechnology has often managed to provide solutions, effective responses, and valid strategies to support the fight against SARS-CoV-2. This work reports a collection of information on nanomaterials that have been used to counter the spread of the SARS-CoV-2 virus. In particular, the objective of this work was to illustrate the strategies that have made it possible to use the particular properties of nanomaterials, for the production of personal protective equipment (DIP) for the defense against the SARS-CoV-2 virus.

**Keywords:** COVID-19; coronaviruses; personal protective equipment; nanomaterials; PPE; SARS-CoV-2

## 1. Introduction

Nanotechnology is one of the sectors of applied science and technology that concerns the control of matter on a dimensional scale in the order of the nanometer [1].

The Recommendation of the European Commission of 18 October 2011 clarifies that "nanomaterial" means a natural, derived, or manufactured material containing particles in the free state, aggregate, or agglomeration, and in which, for at least 50% of the particles in the numerical size distribution, one or more external dimensions are between 1 nm and 100 nm" [2].

By virtue of their very small size, nanomaterials show behaviors and properties that are very different from those of the same material on a macroscopic scale. First, they have a very high ratio between the atoms on the surface and those inside, since they have very large surfaces compared to the volume; moreover, the atoms on the surface, having unsaturated binding sites, are very reactive and excellent catalysts [3,4].

Nanomaterials include nanopowders with all three dimensions smaller than 100 nm (height, width, depth), nanotubes with two smaller dimensions, thin films with one smaller dimension, and finally nanostructured materials having dimensions greater than 100 nm but made from elements with at least one lower dimension [5].

The reduction of dimensions at the nanometric level constitutes an important step forward towards the miniaturization of matter, which no longer behaves as it can usually be observed at the macroscopic level but assumes a new behavior in which the force of gravity has no value, while the van der Waals forces, the surface tension forces and all those forces that concern the atom and the interaction between atoms become important. The explanation lies in the fact that structures with nanometric dimensions are characterized by a number of surface atoms comparable if not higher than the number of atoms in the rest of the structure. In materials of higher dimensions, on the other hand, the number of surface atoms is much lower than the number of internal atoms and this numerical prevalence

determines the behavior of matter perfectly described by the scientific laws of classical physics and chemistry.

From the above, it is clear how nanostructured materials have aroused great interest as their properties differ significantly from all other materials due to the increase in specific area and the quantum effects present in them. These two factors cause a change or can increase properties such as reactivity, mechanical strength, electrical characteristics of the material, and optical and magnetic properties [5].

Nanotechnology is one of the sectors of applied science and technology that concerns the control of matter on a dimensional scale in the order of the nanometer, or one billionth of a meter, and the development of devices on this scale [6].

Nanotechnologies, since their debut in the last decades of the twentieth century, have provided a significant contribution in many fields of science, opening new scenarios in the textile sector [7,8], food [9,10], construction [11,12], electronic components [13,14] and in agriculture [15,16].

Interest was also directed to the use of nanomaterials in the field of environmental prevention and protection. Materials such as carbon nanotubes have proven to be excellent materials for the purification of contaminated water [17,18].

In recent years, there has been a significant and successful use of nanotechnology in the biomedical sciences due to the unique physicochemical properties of nanomaterials, which offer versatile chemical functionalization for the creation of advanced biomedical tools. Nanomedicine consists precisely of the application of nanotechnologies in the medical and pharmacological fields for the diagnosis, treatment, control, and prevention of diseases [19,20].

The field of study of nanomedicine has two related macro-areas: diagnostics [21] and therapy [22–24]. The value of nanotechnologies in medicine lies in their ability to act on a nanometric scale, therefore with dimensions much larger than that of the human cell, thus allowing nanoparticles to move at the same dimensional level as biological processes, paving the way for the so-called target medicine [25,26].

The recent pandemic, which hit our planet with fatal outcomes and a high rate of reproduction and transmission, was caused by the spread of COVID-19, an acronym for the English coronavirus disease, where 19 indicates the year in which the virus was identified for the first time [27,28].

COVID-19 is an acute respiratory disease caused by the virus called SARS-CoV-2 belonging to the coronavirus family [29].

Coronaviruses cause infections in humans and in various animals, they are capable of infecting different species and this "jump of species" takes place thanks to mutations in the genetic heritage of the virus which make it capable of infecting new animal species, including human beings [30,31].

SARS-CoV-2 would appear to have first appeared in the Wuhan seafood market, where live animals were also present, however, the steps regarding transmission to humans are still poorly understood. There are no definitive indications of whether the infection occurred through contact between bats and humans or whether the virus passed from bats to humans through an intermediate host. The intermediate host may have been the pangolin, also known as the "scaly anteater" [32–34].

Due to the high transmissibility of the virus and international travel, the worldwide spread of SARS-CoV-2 increased exponentially during the pandemic, thus posing a serious threat to global public health, so much so that on 11 March 2020 the WHO director- generally officially recognized the outbreak as a pandemic [35].

The following work generally concerns the applications of nanotechnology in the prevention and control of infection caused by SARS-CoV-2. In particular, the aim is to illustrate the strategies and methods by which the unique properties of nano-sized materials are used in the development of PPE. Following an ever-increasing sensitivity that has developed to the problem of the pandemic, a large research activity has led to many publications in the literature. In this work, the attention was focused only on some research which has

proven to be more significant and which allows us to represent a general framework on the application of nanomaterials for the creation of personal protective equipment.

## 2. Nanotechnology in the Fight against SARS-CoV-2

Nanotechnology can provide promising solutions in the fight against the SARS-CoV-2 virus, thanks to the precious peculiarities of the nanoparticles involved such as a high surface-to-volume ratio, easy surface modification, high physic-chemical stability, optical properties specifications, and targeted and controlled release capabilities [36–38]. The potential of nanotechnologies toward SARS-CoV-2 is currently established in prevention, diagnosis, and therapy [39,40] (Figure 1).

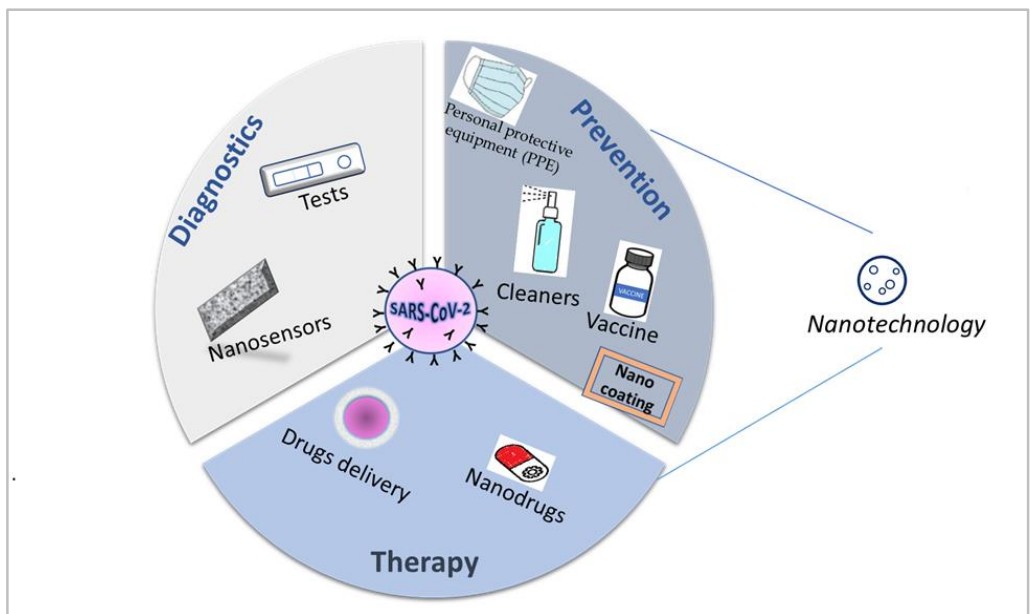

**Figure 1.** Main applications of nanotechnology for the fight against SARS-CoV-2.

Scientists around the world have worked hard to study and develop highly sensitive diagnostic tools to detect the virus quickly and accurately and to replace them with techniques that instead require more time, high costs, and expert users [41–43].

The new sensing tests, based on smart nanometer sensors, are fast, sensitive, and cost-effective, requiring a smaller sample volume and less laboratory equipment. Nanomaterials are used in the design of highly specific and sensitive nanometer sensors for rapid identification of infection. Nanobiosensors are analytical devices comprising a biomarker, a transducer, and a signal amplifier used for the detection of molecules. They have the advantage of selectively detecting all types of analytes by combining the considerable electrical and optical properties of nanomaterials with biological or synthetic molecules employed as receptors [44–46].

Graphene-containing electrochemical biosensors can be used advantageously for the detection of the SARS-CoV-2 virus. The detection mechanism involves the immobilization of specific biomolecules on the surface of the graphene, capable of interacting with the virus, causing a change in the electrical properties of the graphene [47]. To be used in these biosensors, graphene must first be functionalized, creating functional groups on its surface, such as carboxylic acids, amines, and thiols, etc., capable of reacting and therefore immobilizing biomolecules, such as antibodies or nucleic acids, sensitive to SARS-CoV-2 (Figure 2).

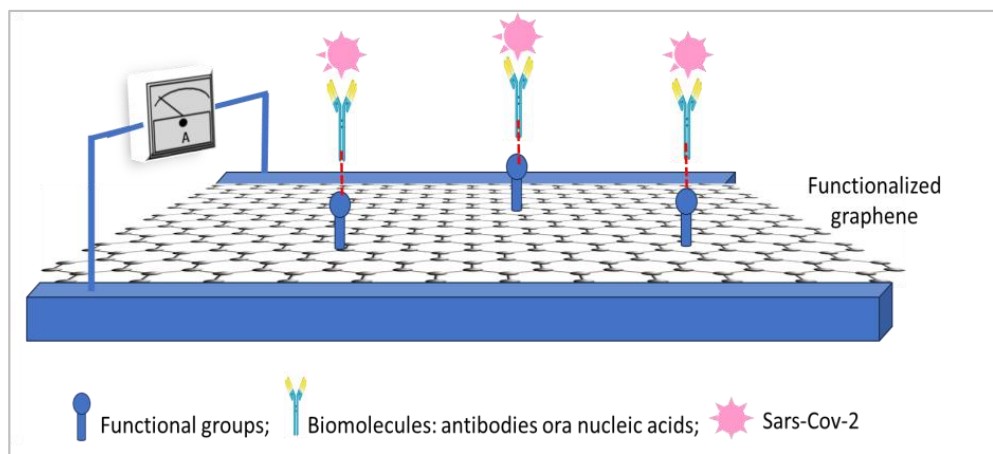

**Figure 2.** Biosensor with functionalized graphene containing biomolecules attracting SARS-CoV-2.

Graphene has shown itself to be very sensitive towards the detection of the virus thanks to its unique properties such as the high surface area and its high conductivity. The use of graphene in biosensors has proved to be more advantageous than other nanomaterials such as gold nanoparticles, silicon being cheaper, and offering greater sensitivity and therefore faster response times. Despite these evident advantages in the use of graphene for the realization of biosensors for the detection of the SARS-CoV-2 virus, an intense research phase is still needed to optimize, for example, the functionalization phases of graphene and more generally a development of more convenient methods of graphene production so that these biosensors can be increasingly cheaper and widely commercialized [48,49].

In general, nano-sized pharmaceuticals exhibit lower toxicities and higher therapeutic efficacy than conventional formulations in the prevention and treatment of viral diseases. The advantages of nanotechnology in antiviral research are many. In fact, in the biomedical field, its use promotes the administration of water-insoluble drugs; improves in vivo drug circulation time, and drug utilization efficiency, and reduces side effects [50,51].

Nanotechnology has recently shown considerable potential in the development of drugs to be used in the treatment of COVID-19 disease [52,53].

In the first months of 2020, numerous research centers in various countries launched experimental pathways aimed at creating safe and effective vaccines against SARS-CoV-2 [54–56].

Although the expected waiting times were longer, some pharmaceutical and biotech companies involved in the trial managed to develop vaccines in less than a year. These vaccines fall into the category of "mRNA drugs" and quickly obtained the approval of the European Medicines Agency (EMA). Vaccines developed by Moderna and Pfizer-BioNTech use messenger ribonucleic acid (mRNA) molecules. The latter is equipped with all the information necessary for the cells to be able to synthesize the Spike protein, present on the external surface of SARS-CoV-2, which is the one that allows the virus access into the host's cells [57].

The Spike protein, produced inside the ribosomes of the host cell, is then expelled and therefore recognized as an antigen by the immune system, which will be able to produce antibodies.

In fact, if the vaccinated person were to later meet SARS-CoV-2, their immune system would recognize it and be ready to fight it. The injected mRNA is encapsulated in lipid nanoparticles (LNPs) to prevent it from being degraded in the bloodstream before reaching the cytoplasm of cells. LNPs are lipids of the size of a few nanometers increasingly used in drug delivery [58].

## 3. PPE with Nanomaterials to Fight the Spread of the SARS-CoV-2 Virus

Nanotechnology is used in the fight against the SARS-CoV-2 pandemic, through the design of new materials capable of inactivating and preventing the spread of the

virus. These nanomaterials have found application above all in the production of personal protective equipment (PPE) such as masks and fabrics.

The FFP1, FFP2, and FFP3 filtering masks are the so-called filtering facepieces (hence the acronym FFP), i.e., respirators with filtering facepieces, defined by current legislation as Personal Protective Equipment (PPE) [59].

They are classified in a very precise manner and subjected to tests that certify their adherence to certain parameters. Unlike surgical masks, they are also highly effective in blocking droplets that remain in the air; breathing, therefore, avoids breathing the droplets of humidity which are the main vector of the virus.

The FFP1, FFP2, and FFP3 masks are required to bear the CE mark; they are produced in compliance with the EN 149-2001 standard which sets the standards of efficiency, breathability, stability of the structure, as well as the technical tests of biocompatibility and the performances of the masks. The progressive numbering (FFP1, FPP2, FFP3) indicates the progressive air filtering capacity of the different types of devices [60].

In a viral epidemic, the protection of healthcare workers and individuals at risk is very important and it is here that nanomaterials can play a fundamental role through their incorporation into personal protective equipment [61,62].

The growing attention that COVID-19 has aroused in the last two years has encouraged and accelerated the design of masks by many researchers, who have worked to modify the surface of the PPE, not only to capture and inactivate the virus, but also to make them washable, reusable, and environmentally friendly, without compromising their effectiveness and safety. For this purpose, various types of nanoparticles have been studied and used, among which the most used are nanoparticles of silver, copper, copper iodide, copper oxide, graphene, graphene oxide, and nanofibres.

### 3.1. PPE with Graphene

Since the beginning of the pandemic, several companies have launched PPE enhanced with graphene microfiber fabrics on the market.

Graphene, as is known, is a nanomaterial consisting of a monoatomic layer of carbon atoms, arranged to form a hexagonal lattice and its structure can be defined as two-dimensional, as there are only two dimensions of the plane [63,64]. By exploiting the antimicrobial, antistatic, and electrically conductive properties of the nanomaterial, researchers have developed face masks with antiviral properties that can be sterilized and reused. We need to dwell on this type of mask.

The new graphene FFP2 produced by the C&S Italy company of Arezzo, called AV Mask Pro [65] would be able to fulfill several functions:

-	repel and block viruses and nanoparticles, ensuring antimicrobial protection, thanks to the inherent antistatic and waterproof capabilities of graphene;
-	reduce the risk of secondary contact infections, caused by the possible presence of bacteria and/or viruses on the external surface of the masks;
-	improve the comfort of those who wear them, guaranteeing lightness, breathability, softness, and resistance.

These masks were made using an original bioactivation process, which allows the integration of graphene within the polypropylene polymer, avoiding the possibility of detachment of nanoparticles, which could occur using more traditional processes such as coating. These masks have passed all tests, and have antiviral activity certified according to ISO 21702:2019 and ISO 18184:2019 standards (Figure 3).

The Abruzzo startup Hygraner managed to patent and create the first antimicrobial, filtering, and breathable non-woven fabric in graphene nanofiber, recyclable and sustainable, with the melt-blown technique [66].

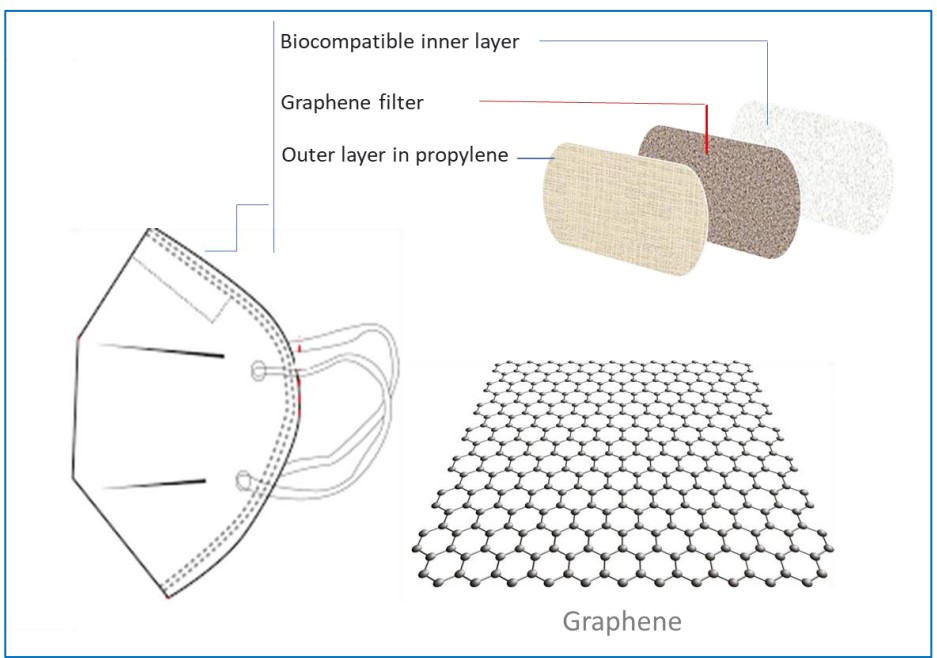

**Figure 3.** Mask with graphene filter.

The fabric is made up of 2 μm diameter polypropylene microfibers, used as an intermediate filtering layer in IR, FFP2, and FF3 surgical-type masks.

With an innovative patent, the company has developed a melt-blown fabric in which fibers are made entirely of a polymer and functionalized graphene, capable of blocking the proliferation of the virus and oxidizing its cellular components, inhibiting all activity.

The fabric, certified according to ISO 18184 which determines the requirements of the antiviral activity of textile products, is hydrophobic, hypoallergenic, antistatic, and non-toxic, soft to the touch, and sustainable for the environment.

Its antiviral property does not decline over time, not even after use, after repeated washing, or other permitted treatments, so the fabric can also be sterilized and reused.

Other research has been conducted for the development of new surgical masks capable of self-sterilization and can be reused or recycled. For these purposes, an original and unique method has been developed to functionalize surgical masks ensuring their self-cleaning and antimicrobial properties [67].

The keystone of the development of this innovative method was the use of the dual-mode laser, which allows depositing an ultra-thin graphene coating of a few layers on a non-woven mask at a low melting temperature (Figure 4).

Specifically, the laser uses a high-energy and precision collimated light beam capable of inducing specific morphological modifications on the surface of the target material. The laser coating allows the so-called graphene-like layers (GL) to be deposited on the support of the mask, which consists of stacked layers of graphene-like nanometric dimensions obtained from a controlled top-down demolition of a target graphite. Generally, to obtain an ultra-thin coating it is possible to use various techniques, many of which involve the use of solvents, which could damage or alter the surface to be coated. The laser technique offers advantageous aspects for the realization of coatings on biomedical materials such as facial masks, thanks to the possibility of controlling various process parameters that could damage the support on which the deposition takes place. In fact, it is not always easy for an implantable material to have all the required characteristics, for example, it can have the right mechanical characteristics but be sensitive to solvents. The laser deposition system can preserve the functionality of the material to be treated by not requiring the use of solvents in direct contact with the surface to be coated.

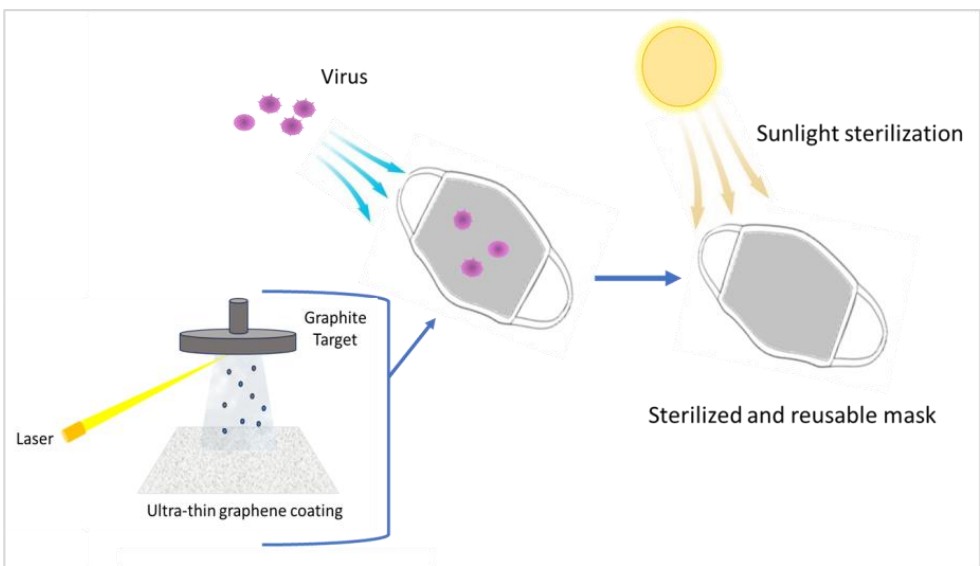

**Figure 4.** The mask, made using an ultra-thin graphene layer deposited by the Laser dual method, is reusable and sterilizable with the aid of solar radiation.

In laboratory tests, the surfaces of these masks were found to be highly hydrophobic and capable of causing incoming water droplets to bounce back. These masks have photothermal properties which give them two important prerogatives: the ability to self-sterilize and virucidal efficacy (inactivation of the virus). The presence of graphene, in fact, determines a photothermal effect, i.e., the production of heat following the absorption of light. The surface temperature of the functional mask, irradiated by the sun's rays, can rapidly rise to over 80 °C, making the masks reusable. As is known, most viruses cannot survive at high temperatures, therefore, the solar radiation to which the surface of the graphene layer is subjected causes it to reach temperatures of up to 80 °C which consequently allow the inactivation of the virus. In particular, the results of such studies found that the antimicrobial efficiency of graphene material, with sunlight sterilization, could improve by 99% within 15–20 min [68].

Recent studies have also tested the effectiveness of graphene oxide (GO). It is a stratified material produced by the oxidation of graphite, but unlike the latter, it is strongly oxygenated, and recent studies have tested its antiviral efficacy [69,70].

To improve the protective properties of the masks, functionalized fabrics have been created, i.e., normal fabrics (cotton and polyurethane), in whose fibers graphene and graphene oxide sheets have been inserted [71]. These studies demonstrated that when cotton or polyurethane fabrics are functionalized with graphene or graphene oxide the infectivity of the fabric towards the SARS-CoV-2 virus is inhibited.

The data reported in the literature show that both graphene and graphene oxide can be advantageously used as materials for the manufacture of personal protective equipment. However, the latter have different characteristics. Graphene oxide, in fact, thanks to the presence on the surface of functional groups such as hydroxyl, epoxy, and carboxylic groups, has a greater hydrophilicity compared to graphene (Figure 5a). Consequently, depending on the final product to be created, graphene can be used to create a more hydrophobic material. Graphene oxide can be more advantageously used for internal layers or more generally in molecular diagnostic devices involving applications in aqueous solutions [72].

Both graphene and graphene oxide can block the virus, preventing its diffusion, facilitated by the different dimensions between the carbon rings, present in both graphene and graphene oxide [73], and the dimensions of the virus [74] (Figure 5b).

Hu et al., reported the creation of a cotton fabric, with antimicrobial properties, on which graphene oxide was inserted and subsequently implanted with $Fe^{3+}$ ions [75].

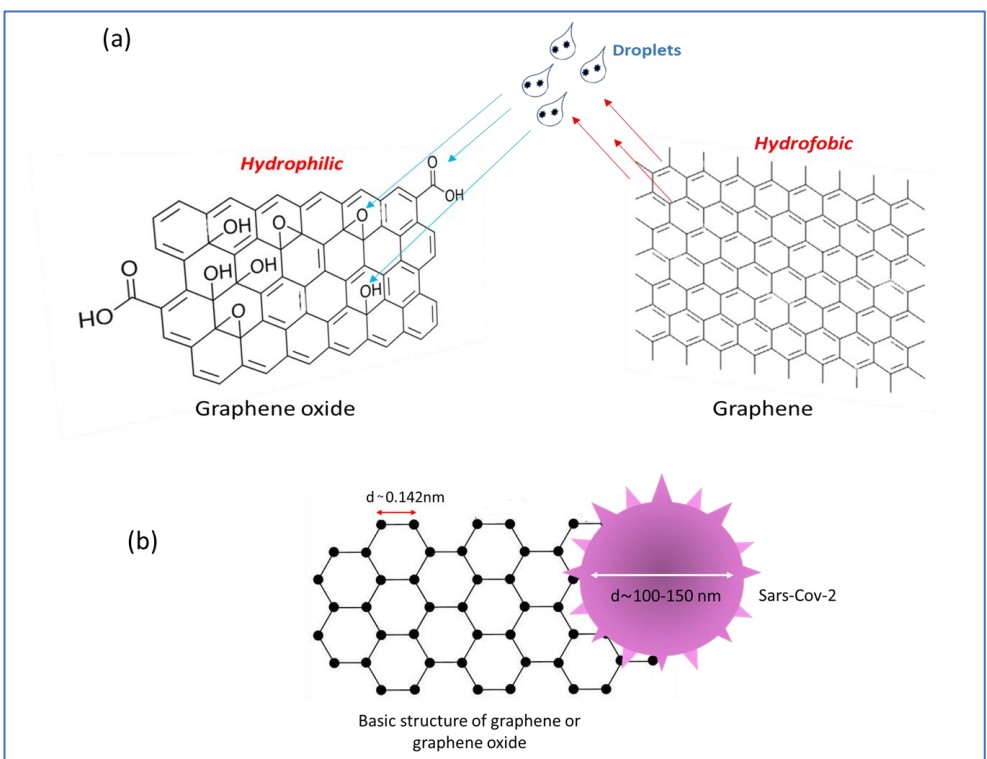

**Figure 5. (a)** Hydrophilic and hydrophobic action, respectively, of graphene oxide and graphene, with respect to droplets; **(b)** size of virus and carbon rings.

Ion implantation is a technique capable of modifying the properties, structure, and morphology of surfaces of carbon-based materials using ions with different energies and doses in a controlled way.

Specifically, cotton fabric was treated with an aqueous dispersion of graphene oxide and placed in a microwave oven for programmed times. The fabric containing graphene oxide, obtained after drying, was subjected to ion implantation of $Fe^{3+}$ using an ion implanter, with programmed ion doses and acceleration voltage.

The antimicrobial tests showed that the cotton fabric, containing graphene oxide and implanted with $Fe^{3+}$ ions had a greater activity than the untreated ones and that increasing the dose of $Fe^{3+}$ showed a greater antimicrobial activity. Although the tests have not been performed strictly on the SARS-CoV-2 virus, this study is important as regards the technique performed and the results of antimicrobial activity that the treated tissues have shown. Given the results obtained, it can be expected that these fabrics can also be efficient for the SARS-CoV-2 virus in the same way and therefore it is to be hoped that there will be an in-depth analysis of the research activities on this study.

The set of information reported in the literature shows that both graphene and graphene oxide can inactivate the SARS-CoV-2 virus, although they use different inactivation mechanisms and therefore, they can be used to make protective devices against the SARS-CoV-2 virus.

Now it is not easy to make a precise comparison between the activity of the two materials as the data reported are carried out under different experimental conditions and therefore further studies are desirable.

### 3.2. Nanoparticles against SARS-CoV-2

As is known, viruses replicate only within a host cell, through different mechanisms that can be summarized as follows:

- attack on the cell membrane;
- penetration of the virus into the cytoplasm of the cell;

- replication exploiting enzymes and organelles of the host cell;
- escape of the virus from the cell.

In general, the antiviral action of nanoparticles mainly intervenes in these interaction mechanisms of the virus with the cell.

Lysenko et al. (2018) [76] proposed that nanoparticles adsorbed on the cell surface induce an alteration of the membrane with consequent contrast to the attachment and penetration of the virus. Most often, nanoparticles work by altering the structure of the capsid protein and reducing viral activity. The capsid of the SARS-CoV-2 virus consists of four structural proteins, known as Spike, Envelope, Membrane, and Nucleocapsid. The large Spike (S) protein, which forms a kind of crown on the surface of the viral particles, acts as a real "anchor" which allows the virus to dock to a receptor expressed on the membrane of the host cells, the receptor ACE2 (Angiotensin-Converting Enzyme 2), allowing the initiation of entry of the virus into the cell. Once penetrated inside, the virus exploits the functional mechanisms of the cell to multiply, which dies, releasing millions of new viruses (Figure 6a). The attack between the Spike protein and the AC2 cell receptor, in the presence of nanoparticles, is prevented with the consequent impossibility of the virus to enter the cell and multiply (Figure 6b).

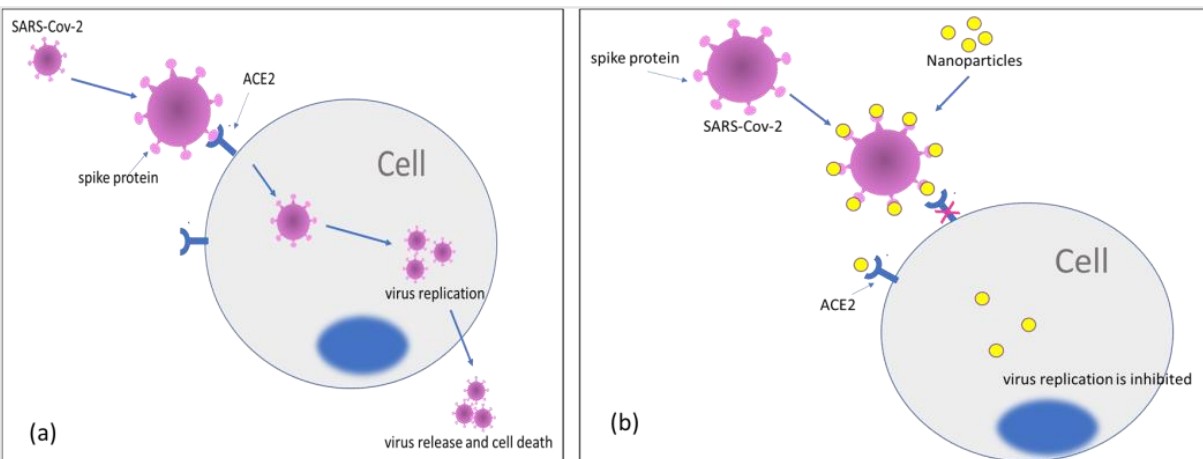

**Figure 6.** Synthetic scheme of the mechanism of attack of the virus to a cell: in the absence (**a**) and in the presence (**b**) of nanoparticles.

3.2.1. PPE with Silver Nanoparticles

Since ancient times, such as the time of the Greeks and Romans, products based on $Ag^+$ ions, known for their antimicrobial properties, were used as medicines and for water conservation [77].

The antibacterial action of silver depends on the biologically active silver ion ($Ag^+$), capable of irreversibly damaging the key system of enzymes in the membrane of pathogens.

Silver nanoparticles (AgNPs) have a size between 1 nm and 100 nm; although they are often described as silver, some of them are composed, in large percentages, of silver oxide, due to the high ratio between surface and volume.

AgNPs exhibit physicochemical characteristics, as their high surface/volume ratio allows a great synergy with microbial surfaces, which leads to high antimicrobial safety [78]. The size and shape of the AgNPs play a determining role; their size, in fact, allows a constant release of ions having antimicrobial activity, unlike salts which rapidly release the ions, quickly wearing off this effect.

With the use of antibiotics, silver-based disinfectants have become obsolete, surviving only in traditional ethnic medicine, but in recent years they have made a comeback because they are used in cases of antibiotic resistance, which leave the sick defenseless against numerous infections. Colloidal silver has proved to be a very valid remedy for the skin, both to ward off dangerous infections in burn victims and to reactivate the metabolism

of damaged skin tissues. Colloidal silver, being made up of nanometric particles of silver suspended in totally purified water, is a liquid compound.

Today, silver is considered one of the most powerful natural disinfectants [79].

Several companies that are at the forefront of nanotechnology have produced innovative masks to fight the SARS-CoV-2 virus.

It is appropriate to dwell on some of the innovative masks made recently containing silver nanoparticles. A team of researchers from the National Autonomous University of Mexico (UNAM), led by researcher Sandra Rodil, has created a face mask called "SakCu" (in the Mayan language "Sak" means silver, while "Cu" indicates the symbol of copper) able not only to protect against COVID-19 but also to inactivate the virus [80,81].

The mask consists of three layers: the central one is composed of silver-copper nanolayers deposited in polypropylene, while the first and third are made of cotton. The researchers thus used a silver-copper mixture, which forms a nanolayer between 30 and 40 nanometers thick, which ensures double protection against viruses and bacteria.

To test the mask, the researchers used a few drops contaminated with the virus and placed them on the silver-copper film deposited on polypropylene.

The result was satisfactory, since when the viral concentration was high, the virus disappeared by more than 80% in about 8 h; however, when the viral load was low, no virus RNA was detected within 2 h. The researchers, illustrating their discovery, concluded that the virus RNA was damaged by contact with the Ag-Cu nanolayer.

Since the surface of the mask does not remain contaminated, unlike many others, its disposal would not create any problems, moreover, the mask is reusable and washable up to ten times without losing its properties. Based on this research, the Kolzer Company of Cologno Monzese has given its availability to use the technology it possesses in the production of masks with an antibacterial, water-repellent, waterproof, ionizing, and temperature-regulating device [82]. The NANOX biotechnology company, supported by the FAPESP (Program Innovative Research in Small Enterprises) of São Paulo of Brazil, has produced a particular fabric impregnated on the surface with silver nanoparticles, capable of inactivating the SARS-CoV-2 virus through the simple contact.

In tests carried out in the laboratory, it was found, in fact, that it took only two minutes, after contact, to eliminate 99.9% of the virus. This extraordinary tissue was developed by an international research team led by Brazilian researchers from the Institute of Biomedical Sciences of the University of São Paulo (ICB-USP).

The fabric is made from a blend of polyester and cotton (polycotton) containing silver nanoparticles soaked into the surface through a process of dipping, followed by drying and fixing, called "dry curing" [83].

The laboratory tests analyzed several tissue samples, with and without embedded silver nanoparticles, in test tubes with a solution containing large amounts of cellular SARS-CoV-2. All samples were kept in direct contact with the viruses at different time intervals to evaluate their antiviral activity.

The results of the analyzes were satisfactory as they established that the tissue samples, containing silver nanoparticles, inactivated 99.9% of the virus after a few minutes of contact.

### 3.2.2. PPE with Copper Nanoparticles

Copper is bacteriostatic, i.e., it fights the proliferation of bacteria on its surface. In the health sector, it has always been used for hospital door handles, IV poles, and all those accessories that are touched, and which can constitute a vehicle for contagion.

Copper works as a barrier for bacteria and viruses, as it contains positively charged ions that trap viruses, which are instead negatively charged and so the positive copper ions penetrate the viruses, preventing them from replicating. The characteristics of copper are comparable to those of other much more expensive metals with antibacterial activity, such as silver and gold [84].

Copper nanomaterials are an emerging class of nano-antimicrobials that provide complementary effects and characteristics, compared to other nano-sized metals, such as silver or zinc oxide nanoparticles.

It is established that soluble copper compounds provide excellent antimicrobial activity against bacteria, fungi, algae, and viruses. It appears that copper can generate reactive hydroxyl radicals, which can cause irreparable damage such as oxidation of proteins, cleavage of DNA and RNA molecules, and membrane damage due to lipid peroxidation [85].

The antibacterial and antiviral activity of copper has been known since ancient times. Its antimicrobial properties were known in ancient Egypt, as the metal was used for the sterilization of water and the treatment of wounds, as reported in medical texts dating back to around 2500 BC. As was the case with silver, also for copper the advent of antibiotics in 1932 determined its eclipse [53].

Some nanotechnology experts have invested their efforts and expertise in the development of fabrics treated with copper nanoparticles, to prevent the spread and replication of the virus.

Firmly convinced of the bacteriostatic and antiviral properties of copper, researchers have developed different types of hypoallergenic masks that offer a high level of protection combined with comfort and environmental sustainability.

A textile company in the Como area, Italtex SpA in partnership with the Ambrofibre company in Milan, has patented "Virkill", the registered trademark of the first fabric made in Italy with copper nanoparticles that are very effective against bacteria and viruses [86].

Colored between yellow and orange, Virkill has the consistency of a normal fabric, but the element that makes it innovative is the insertion, with an industrial process, of "fused" copper nanoparticles in the thread. This fabric is suitable for its use in the PPE sectors against the propagation of viruses such as SARS-CoV-2.

It should be emphasized that the copper nanoparticles are not deposited on the surface of the fabric but are inserted into the polymer of the thread before its formation.

The copper nanoparticles, inserted inside the thread, preserve the surface of the fabric from contamination by viruses and bacteria, as certified by the ISO 18184:2019 tests for the determination of its antiviral activity against the SARS-CoV-2 virus, and by UNI EN ISO 20743: 2013 for the determination of its antibacterial activity. In particular, the test found an antiviral activity that corresponds to a virus inactivation of more than 99.9%.

The presence of nanoparticles inside the polymer ensures a high resistance to washing, in water and dry, so the antiviral property of Virkill is long-lasting.

Other appreciable characteristics of the fabric are breathability and compatibility in contact with the wearer's skin.

The high breathability of Virkill has been confirmed by the results of tests compliant with the UNI EN ISO 11092:2014 standard.

The second peculiarity was tested with positive results by the patch test, an exam used for the diagnosis of allergic contact reactions, resulting in a hypoallergenic fabric.

### 3.2.3. PPE with Copper Iodide Nanoparticles

Copper iodide particles have proven to be potent microbicidal agents due to their high surface area-to-volume ratio. A recent study reports the deposition of copper iodide nanoparticles on cotton fabric with an ultrasound method [87].

This ultrasonic coating is free of toxic materials and is a process that has a significant practical advantage as it is fast, simple, economical, and environmentally friendly by involving the principles of "green chemistry".

In particular, the study concerned the antimicrobial activity of natural cotton coated with copper iodide and covered with Hibiscus rosa-sinensis flower extract (CuI-FE) rich in anthocyanin and cyanidin 3-soforoside.

As is known, copper iodide (CuI) is an inorganic chemical compound, while Hibiscus rosa sinensis is a shrub belonging to the Malvaceae family, originally from China.

Since CuI particles exhibit good antibacterial and antitumor activity, a "molecular docking" study was conducted with CuI-FE synthesized to act against the COVID-19 main protease protein. It should be remembered that "molecular docking" is a method capable of predicting the preferred orientation of one molecule towards another, favoring their bond and therefore the formation of a stable complex. The synthesized CuI-FE, which was ultrasonically deposited on natural cotton to improve its multifunctional properties, was well adsorbed on the cotton surface due to physical and chemical interactions. Cotton coated with CuI-FE, showing good antibacterial activity against infections caused by broad-spectrum bacteria and against the main protease of the coronavirus, answers the need for an efficient material to counter the SARS-CoV-2 virus.

### 3.2.4. PPE Containing Copper Oxide Nanoparticles

A study conducted in 2018 by Redwanul Hasan (Bangladesh University of Textiles) dealt with the development of antimicrobial cotton fabrics, using copper oxide nanoparticles applied directly to the fabric, using the pad-dry-cure method [88].

This method consists in preparing a suspension of NPs in an aqueous solution containing both stabilizing agents, essential to prevent the particles from settling, and fixing or binding agents, which bind the particles to the fibers. At this point the tissue is immersed in the suspension for a certain period, then it is extracted, sterilized, dried, and subjected to curing treatments and possible washings (Figure 7).

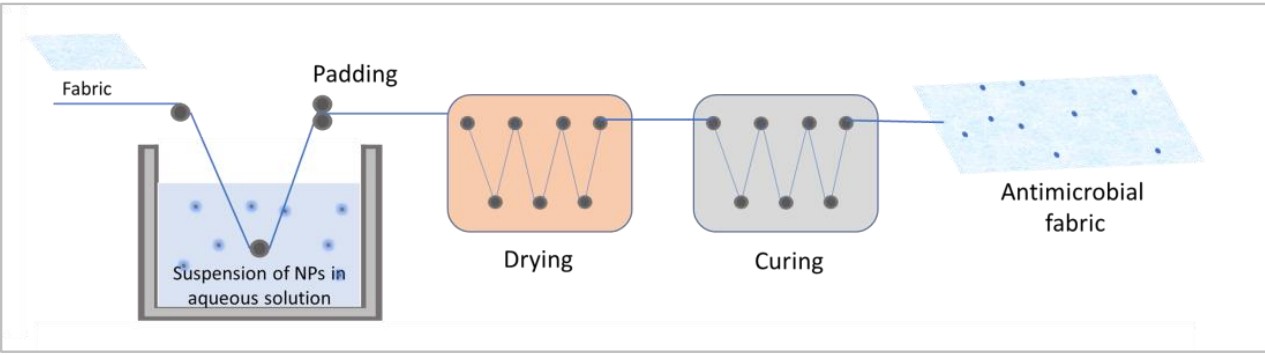

**Figure 7.** Pad–dry–cure method.

The method, which is quite simple, however, has some drawbacks because it requires various steps which involve manual work and the use of time, but above all heating steps with high temperatures imply considerable energy consumption.

In laboratory tests, treated and untreated fabrics were compared, and the results obtained found the incorporation of copper nanoparticles into the treated fabrics, as well as a significant antibacterial activity. Another test showed that the treated fabric can withstand up to 25 washing cycles This technology, in addition to cotton, can be extended to polyester, silk, and other fabrics to make PPE against SARS-CoV-2.

### 3.2.5. PPE with Zinc Oxide

Zinc oxide nanoparticles exhibit well-known antimicrobial properties [89,90].

They represent a possible alternative, albeit with a less powerful antimicrobial action, to other metals such as silver and copper, being cheaper and easier to find. The antiviral action is determined by the action of the zinc ions which prevent the binding of the virus with the host cell thus leading to the deactivation of the virus [91].

A recent study reports a method in which zinc oxide nanoparticles are synthesized directly into textile materials which can be used for the creation of PPE [92]. This method can potentially be applied to both natural and synthetic fibers and is called "Crescoating". It consists of the growth of zinc nanoparticles in situ starting from an initial solution in which the dissolved ionic precursor is present. The fabric is soaked in the initial solution

and subsequently subjected to heat treatment to trigger the nucleation of the nanoparticles on the fabric. The results showed that the particles are strongly anchored to the fabric resulting in greater stability and durability compared to other surface-treated materials. The antiviral properties of the fabric show a reduction of the virus of 99.99% before washing, after 10 min of contact with the virus, and an efficacy of 99.8% after 50 washing cycles, demonstrating excellent adhesion of the nanoparticles and an important reuse of fabric. The "Crescoating" method is, therefore, to be considered valid for the creation of masks that are cheap, durable, reusable, and effective against the SARS-CoV-2 virus.

### 3.2.6. PPE with Titanium Oxide

Titanium oxide ($TiO_2$) is known to be a readily available photocatalytic material. In general, the mechanism of antimicrobial action of titanium oxide can be attributed to the production of reactive oxygen species (ROS) which lead to an alteration of the virus proteins or damage to the DNA [93] The use of titanium oxide nanoparticles represents an excellent strategy to increase contrast actions against SARS-CoV-2. These nanoparticles, therefore, through the combined action with light can be used to develop self-decontaminating materials capable of inactivating the virus. The researchers of the Coraero project, in collaboration with the University of Milan-Bicocca, have highlighted, through experimental tests and simulations, that the titanium oxide nanoparticles, deposited on the surfaces, can both adsorb the virus, but also make it inactive through a combined action of light or thermal treatments [94].

The adsorption phase of the virus on the surface of the titanium oxide nanoparticles is due to the interaction of the latter with the Spike protein of the virus. In particular, in the amino acids that make up the protein, there are nitrogen, sulfur, and oxygen atoms which, with their electronic doublets, can interact with the titanium atom which acts as a Lewis acid. The functional groups $-NH_2$, -SH, -OH, -COOH present in the spike protein, therefore interacting with the surface of the titanium oxide, favor the adsorption of the viral particles. As regards the deactivation of the SARS-CoV-2 virus following exposure to UV light, the mechanisms of action of titanium oxide nanoparticles are of three types: (I) denaturation of the viral protein; (II) damage to the viral genome by absorption of UV radiation by the nucleic acid; (III) photocatalytic oxidation of the virus. The deactivation of the virus through thermal treatments, on the other hand, is caused by the separation of the proteins, adsorbed by the $TiO_2$ nanoparticles, from the virus structure with consequent inactivation of the virus. The adsorption and inactivation activity of photosensitive titanium oxide nanoparticles is particularly interesting when talking about their possible use in the production of personal protective equipment such as masks, gowns, etc.

Most of the masks available on the market are made of polyester, polyamide, or synthetic non-woven fibers containing titanium oxide. However, it is important to specify that studies are still needed to identify reliable methods of analysis to evaluate the release of particles and estimate their exposure.

A 2021 study by scientists from the Belgian Institute of Public Health, Sciensano, highlighted the presence of titanium dioxide nanoparticles in the synthetic fibers of masks sold to the public with concentrations from a few milligrams to 150 mg per mask [95].

It should be noted that according to the regulation on classification, labeling, and packaging (CLP) titanium dioxide particles with a diameter less than or equal to 10 μm in a concentration greater than 0.1% are designated as suspected carcinogenic by inhalation. Considering prolonged and intensive use of the masks, the value of 3.6μg was evaluated as the limit fraction of particles that must be released from the fibers of the mask to exceed the acceptable exposure level [96].

### 3.3. PPE with Nanofibers

Nanofiber fabrics consist of filaments of polymer materials of nanometric dimensions [97]. Nanofibers are usually not used as a single strand but in the form of a messy weave with a non-woven structure. Nanofibers, thanks to a very small diameter compared

to other natural fibers or microfibres (Table 1), have a high surface/volume ratio and a high specific surface area (lateral area/weight). These fabrics, being generally constituted by polymeric nanofibres, are hydrophobic and highly flexible [98,99].

**Table 1.** Diameter of different fibers.

| Fiber | Diameter (µm) |
| --- | --- |
| Wool | 30–120 |
| Silk | 20 |
| Microfiber | 2–5 |
| Nanofiber | 0.05–1 |

Nanofibers can be produced by electrospinning. The process is based on the application of high voltages to a flow of a polymer solution. The polymeric jet during its flight towards the manifold stretches and becomes thinner. Following the evaporation of the solvent, the nanofibers are deposited on a substrate (Figure 8).

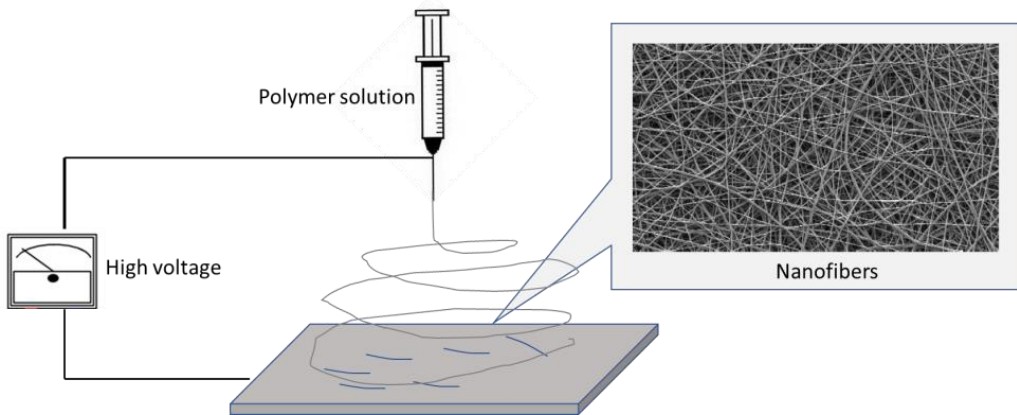

**Figure 8.** Electrospinning nanofibers at high voltage.

These nanofibers, therefore, have perfect characteristics for multiple areas of application, including the creation of high-performance filter masks [100].

Since SARS-CoV-2 spreads mainly through respiratory droplets produced and by close contact between infected people, the use of appropriate PPE, with novel functions such as hydrophobicity and antimicrobial activity, without compromising the consistency or breathability of the material used is a desirable expectation.

The hydrophobicity of PPE products alone can provide an effective barrier against airborne droplets emitted when you cough or sneeze.

Traditional face masks have a gap between the fibers, on average 10–30 µm, which is not adequate to avoid contact with the virus. A nanofiber fabric has a marked porosity and small pores.

In principle, a face mask should be comfortable and effective. Nanofibers for making masks respond well to these needs. The latter significantly reduces, compared to traditional filters, the resistance of the airflow, with a consequent improvement in breathability. Therefore, the use of nanofibers for the production of face masks or respiratory filters increases respiratory comfort as well as having high filtering properties.

The above-mentioned ensures far superior protection compared to traditional surgical masks which do not protect against particles of size between 10 and 80 nm.

Nanofiber fabrics can therefore be used advantageously for the creation of masks thanks to a double action: water repellency towards the droplets that can carry the virus and blocking the action of the virus thanks to a structure made up of pores of smaller dimensions than those of the virus (Figure 9)

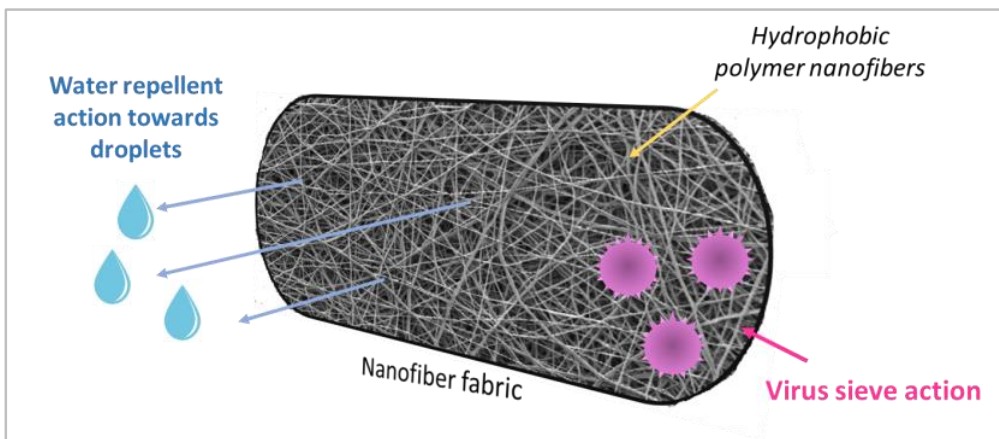

**Figure 9.** The double action of nanofiber fabrics: water repellency towards droplets and sieving and blocking of the virus.

As is known, air transport for COVID-19 consists of tiny droplets and particulate matter of infected people and environmental aerosols present in the air. These poly-dispersed aerosols are of different sizes and can interact with each other by interfering with the filtration process. Chinese scientists, Leung and Sun, for the first time studied the effects of the shape and size of environmental aerosols that simulate the bio-nano aerosols of the coronavirus, on the efficiency of the nanofibers filters [101]. Tests were performed on electrostatically charged PVDF nanofiber filters and manufactured as multiple modules of 2, 4, and 6, with each module having 0.765 g/m$^2$ of electrostatically charged PVDF nanofibers. The authors reported that a PVDF filter, either charged or uncharged, can capture ambient aerosols with sizes ranging from 10 to 400 nm. When the nanofibers are charged, dielectrophoresis improves the capture of aerosols larger than 80 nm. A nanofiber filter charged and made up of 6 modules can achieve an efficiency of 88%, 88%, and 96% for the ambiental aerosol size of 50, 100, and 300 nm respectively.

Nanomaterials such as nanofibres, used as components of high-performance filter masks (FFP2, FFP3), minimize the dispersion of respiratory droplets, released by infected subjects in the form of aerosols. General characteristics of anti-SARS-CoV-2 masks prepared with different nanomaterials are reported in Table 2.

**Table 2.** General characteristics of anti-SARS-CoV-2 masks prepared with different nanomaterials.

| Nanomaterials | Classification | General Characteristics |
|---|---|---|
| Graphene [60–65] | | <ul><li>Ability to block the virus thanks to the pore size of the graphene structure smaller than the size of the SARS-CoV-2 virus.</li><li>Water-repellent properties towards droplets.</li><li>Photothermal effect.</li><li>Sunlight sterilization.</li><li>Maintains the filtering, antiviral capacity of the mask intact after repeated washing.</li><li>The graphene layer is inserted in layers of a polymeric and patibiocompatible nature.</li></ul> |

**Table 2.** *Cont.*

| Nanomaterials | Classification | General Characteristics |
|---|---|---|
| Graphene oxide (GO) [66–71] | | • Ability to block the virus, thanks to the size of the pores of the graphene oxide structure smaller than the size of the SARS-CoV-2 virus.<br>• Hydrophilic properties, cannot repel droplets.<br>• Mainly used for the inner layers of masks.<br>• The graphene oxide layer is inserted into layers of polymeric and bio-combustible nature. |
| Nanoparticles [73–85] | • Silver nanoparticles (AgNPs)<br>• Copper nanoparticles (CuNPs)<br>• Copper iodide nanoparticles (CuINPs)<br>• Copper oxide (CuONPs)<br>• Zinc oxide nanoparticles (ZnONPs) | • Effectiveness up to 99.9% in deactivating the SARS-CoV-2 virus.<br>• Nanoparticles are deposited or inserted on tissue or polymeric supports.<br>• The nanoparticles act by releasing ions that interact with the genetic material of the virus, preventing its replication. |
| [93,94] | • Titanium oxide (TiO$_2$NPs) | • Nanoparticles are deposited or inserted on tissue or polymeric supports.<br>• High antiviral efficiency with adsorption and photo inactivation of the virus |
| Nanofibers [90–94] | • Polymeric nanofibers<br>• PVDF (Polyvinylidene fluoride Polyvinylidene fluoride) | • High antiviral efficiency with 99.9% virus-blocking ability.<br>• The high breathability of the masks.<br>• Low cost of preparation. |

## 4. Conclusions

The work of prevention and control of the recent pandemic has made use not only of previous experiences and knowledge acquired in the management of previous epidemics but also of the progress made by nanotechnologies in the design and implementation of feasible solutions to prevent and contain the infection.

Reusable, washable, and sustainable masks have been developed that are able not only to protect against the virus but also to inactivate it as some of them have nanolayers inside them that damage the viral RNA; others are covered in a water-repellent fabric that releases metal ions capable of repelling droplets in the air that deposit on the mask. These masks, even if they cost more than traditional ones, have the advantage of providing effective prevention of contagion and reducing the environmental impact caused by disposable masks.

In this period of greater sensitivity toward the problems created by the recent pandemic, this research sector is very active and highly prolific in publications.

Therefore, this work aimed to identify only some of the most significant results reported in the literature and on the market to allow a broad and general overview of the application of nanomaterials for the creation of PPE.

The various nanomaterials reported in this review first of all show that they are all capable of contributing in a positive and effective way to countering the spread of the SARS-CoV-2 virus, although their way of acting, their characteristics are often different.

Graphene, graphene oxide, and nanofibers mainly act as real filters capable of blocking the virus, thanks to the size of the pores present in their structures which are smaller than the size of the SARS-CoV-2 virus. So in this respect, it is more precise to talk about blocking the virus.

Metallic nanoparticles act differently, releasing ions that come into contact with the genetic material of the virus, thus leading to its inactivation. So in the case of metal nanoparticles, it is more correct to speak of inactivation of the virus. Beyond this substantial difference in the mechanism of action, there are differences even within these two categories of nanomaterials.

Graphene and polymeric nanofibres, in general, have a water-repellent behavior, and therefore exhibit an action of removing droplets, which, as is known, are small water

droplets and are among the most common carriers of the virus. These materials, therefore, have a double action: filtering of the virus that manages to reach the surface and preventive action against droplets.

Graphene oxide, on the other hand, has a hydrophilic nature, therefore it has no action to contrast droplets but only a virus-blocking action. In fact, in the creation of masks, their use is mainly intended for internal layers.

Graphene has another important characteristic that of the photothermic effect, i.e., if subjected to sunlight it heats up leading to the death of the virus.

Graphene, therefore, presents itself, under this aspect, as one of the best materials exhibiting virus-blocking properties, preventive droplet contrast, and virus inactivation after exposure to sunlight. This means that the graphene masks could potentially be reused after sterilization in sunlight.

As far as nanoparticles are concerned, almost all of them have a high virus deactivation efficacy which can reach up to 99.9%. The data reported in the literature still do not allow for the creation of a precise scale of efficacy as the laboratory tests are not yet standardized. However, it is possible to say that the effectiveness of nanoparticles depends on their size, shape, nanoparticle/support ratio, and the construction characteristics of the nanomaterial.

Other important features of the face masks against the SARS-CoV-2 virus, in addition to effectiveness, are comfort, breathability, and safety.

Safety is a fundamental aspect, but to date with respect to nanomaterials there is a scarcity of information against a high level of pervasiveness on the market. These gaps are to be attributed to a certain fragmentation of the studies available so far in the literature, which does not find a real common line.

A common fear is that the nanomaterials inserted into the masks could be released and meet the human body [102,103].

In most cases, however, manufacturers ensure that the insertion of nanomaterials inside the masks has been carried out with technologies, often covered by patent secrets, capable of preventing any release harmful to human health.

Most of the products put on the market, according to the declarations of the producers, are tested and certified by independent bodies, while for the products in the research phase, the tests are carried out as a preliminary phase, in the laboratory.

Comparing the materials presented, with respect to the comfort and breathability of the masks, an important position is occupied by nanofibres. In fact, they themselves form a nanofibre membrane that allows high breathability, flexibility, and therefore high comfort for the wearer. For other materials, support is always necessary, such as synthetic or natural fabric, in which to insert the nanomaterials which, depending on their nature, can interfere with the breathability and comfort of the mask.

According to the European Chemicals Agency (ECHA), nanotechnologies will be increasingly present in daily life and this sector is expected to develop very rapidly which will make a significant contribution to industrial and economic growth in the coming years. According to widespread scientific opinion, nanotechnologies will go through 4 stages of development. The first two are already underway and extensively developed. The first concerns the use of nanomaterials in "passive nanostructures" which can be the creation of nanoreinforced nanomaterials or the application of nanocoatings that can be used for example for the creation of self-cleaning or antimicrobial surfaces.

The second phase, which has already begun, is aimed above all at what could be defined as "active nanostructures", i.e., capable, thanks to their bioactivity, of being used as target drugs, for example by coating the nanoparticles with specific proteins capable of interacting with the virus, the bacterium, the organ or with the diseased cell. The third and fourth phases will be more complex and highly sophisticated and will concern the creation of nanosystems to then move on to the generation of molecular nanorobotics capable, for example, of using energy and promoting a series of specific functions such as intercepting and virus elimination.

Current research has clearly shown the important role that nanomaterials, as well as nanoparticles, can play in the fight against viral diseases and especially against SARS-CoV-2. Nanotechnology paves the way for the research of new drugs and treatments against COVID-19 by addressing the problems of low biocompatibility, poor stability, and toxicity issues.

Nanotechnology applications, however, present some critical issues that need to be addressed to facilitate their wider implementation in the healthcare field. It is important to underline that particular attention must be paid to the behavior of nanomaterials when they come into contact with the human body, for example, inhaled or entering the bloodstream, where they can potentially change their behavior. Most of the studies reported in the literature have evaluated biocompatibility only through laboratory and in vitro tests and using generic protocols. Precise standardized protocols for characterizing the properties of nanomaterials, with particular reference to their behavior with respect to human health, are desirable.

Overcoming these challenges requires effective collaboration between experts in materials science, pharmacology, toxicology, and regulatory agencies.

Another obstacle to overcome is the production of nanomaterials on a large scale in order to make possible feasible and economic commercialization of nanoparticle-based products.

In conclusion, it can be said that nanotechnologies have already amply highlighted the potential ability to improve various aspects that are crucial for the fight against SARS-CoV-2, such as diagnostics, therapies, and protection.

**Author Contributions:** Conceptualization, P.D.L. and A.M.; methodology, P.D.L. and A.M.; writing—review and editing, P.D.L. and A.M.; supervision, P.D.L., A.M. and J.B.N. All authors have read and agreed to the published version of the manuscript.

**Funding:** This research received no external funding.

**Conflicts of Interest:** The authors declare no conflict of interest.

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
