# Peer review of "Nanomaterials Used in the Preparation of Personal Protective Equipment (PPE) in the Fight against SARS-CoV-2"

_inorganics, doi:10.3390/inorganics11070294_

Round 1

Reviewer 1 Report

(1)    The manuscript mentions a variety of antiviral strategies with different materials. It is necessary to compare their difference such as the antiviral properties.

(2)    The paragraph structure of lines 71,524 and 538 needs to be revised.

(3)    The title of the article focuses on anti-SARS-COV-2. However, the anti-SARS-CoV-2 performance of some materials is not clearly demonstrated and relevant data need to be added to further explain the anti-SARS-CoV-2 performance of materials.

(4)    There is a lack of more perspective on the prospects of nanomaterials in fighting viruses. There needs to be more discussion about the issues that will arise in these areas.

(5)    A table or TOC is highly recommended to summarize the main points in this review.

Moderate editing of English language required

Author Response

Dear Reviewer,

thank you for your time spent reviewing our manuscript and for giving us important tips for improving it. The manuscript has been revised in many parts. The changes have been highlighted in yellow. Here are the answers to your suggestions point by point. We hope that in this form the manuscript will find your approval to be considered for its publication.

We thank you and send you our best regards.

The authors

Comments and Suggestions for Authors

Comments and Suggestions for Authors

  • The manuscript mentions a variety of antiviral strategies with different materials. It is necessary to compare their difference such as the antiviral properties.

1.R) The following part has been added in the text (Lines 638-686):

“The various nanomaterials reported in this review first of all show that they are all capable of contributing in a positive and effective way to countering the spread of the Sars-Cov-2 virus, although their way of acting, their characteristics are often different.

Graphene, graphene oxide and nanofibers mainly act as real filters capable of blocking the virus, thanks to the size of the pores present in their structures which are smaller than the size of the Sars-Cov-2 virus. So, in this respect it is more precise to talk about blocking the virus.

Metallic nanoparticles act differently, releasing ions that meet the genetic material of the virus, thus leading to its inactivation. So in the case of metal nanoparticles it is more correct to speak of inactivation of the virus. Beyond this substantial difference in the mechanism of action, there are differences even within these two categories of nanomaterials.

Graphene and polymeric nanofibres, in general, have a water-repellent behavior, and therefore exhibit an action of removing droplets, which, as is known, are small water droplets and are among the most common carriers of the virus. These materials therefore have a double action: filtering of the virus that manages to reach the surface and preventive action against droplets.

Graphene oxide, on the other hand, has a hydrophilic nature, therefore it has no action to contrast droplets but only a virus blocking action. In fact, in the creation of masks, its use is mainly intended for internal layers.

Graphene has another important characteristic that of the photothermic effect, i.e. if subjected to sunlight it heats up leading to the death of the virus.

Graphene therefore presents itself, under this aspect, as one of the best materials exhibiting virus blocking properties, preventive droplet contrast and virus inactivation after exposure to sunlight. This means that the graphene masks could potentially be reused after sterilization in sunlight.

As far as nanoparticles are concerned, almost all of them have a high virus deactivation efficacy which can reach up to 99.9%. The data reported in the literature still do not allow for the creation of a precise scale of efficacy as the laboratory tests are not yet standardized. However, it is possible to say that the effectiveness of nanoparticles depends on their size, shape, nanoparticle/support ratio and the construction characteristics of the nanomaterial.

Other important features of the face masks against the Sars-Cov-2 virus, in addition to effectiveness, are comfort, breathability and safety.

Safety is a fundamental aspect, but to date with respect to nanomaterials there is a scarcity of information against a high level of pervasiveness on the market. These gaps are to be attributed to a certain fragmentation of the studies available so far in the literature, which do not find a real common line.

A common fear is that the nanomaterials inserted into the masks could be released and meet the human body [98,99].

In most cases, however, manufacturers ensure that the insertion of nanomaterials inside the masks has been carried out with technologies, often covered by patent se-crets, capable of preventing any release harmful to human health.

Comparing the materials presented, with respect to the comfort and breathability of the masks, an important position is occupied by nanofibres. In fact, they themselves form a nanofibre membrane that allows high breathability, flexibility, and therefore high comfort for the wearer. For other materials, a support is always necessary, such as synthetic or natural fabric, in which to insert the nanomaterials which, depending on their nature, can interfere with the breathability and comfort of the mask.

  • The paragraph structure of lines 71,524 and 538 needs to be revised.
  1. Reply)

-Lines 71. The previous sentence has been replaced with:

“The value of nanotechnologies in medicine lies in their ability to act on a nanometric scale, therefore with dimensions much larger than that of the human cell, thus allowing nanoparticles to move at the same dimensional level as biological processes, paving the way for the so-called target medicine [25,26].” (Now lines 71-74)

-Lines 524. The previous sentence has been replaced with:

 “In principle, a face mask should be comfortable and effective. Nanofibers for making masks respond well to these needs. The latter significantly reduce, compared to tradi-tional filters, the resistance of the air flow, with a consequent improvement in breath-ability. Therefore, the use of nanofibers for the production of face masks or respiratory filters increases the respiratory comfort as well as having high filtering properties.” (Now lines 580-584)

- Lines 538: The previous sentence has been replaced with:

“As is known, air transport for COVID-19 consists of tiny droplets and particulate matter of infected people and environmental aerosols present in the air. These polydispersed aerosols are of different sizes and can interact with each other by interfering with the filtration process. Chinese scientists, Leung and Sun, for the first time studied the effects of the shape and size of environmental aerosols that simulate the bio-nanoaerosols of the coronavirus, on the efficiency of in nanofibers filters [97]. Tests were performed on electrostatically charged PVDF nanofiber filters and manufactured as multiple modules of 2, 4 and 6, with each module having 0.765 g/m2 of electrostatically charged PVDF nanofibers. The authors reported that a PVDF filter, either charged or uncharged, can capture ambient aerosols with sizes ranging from 10 to 400 nm. When the nanofibers are charged, dielectrophoresis improves the capture of aerosols larger than 80nm. A nanofiber filter charged and made up of 6 modules can achieve an efficiency of 88%, 88% and 96% for the ambiental aerosol size of 50, 100 and 300 nm respectively.”(Now lines 597-609).

  • The title of the article focuses on anti-SARS-COV-2. However, the anti-SARS-CoV-2 performance of some materials is not clearly demonstrated and relevant data need to be added to further explain the anti-SARS-CoV-2 performance of materials.

3.Replay) see point 1.Replay)

3.1Replay) The following part has been added in the manuscript (Lines 128-151- including new figure 2):

“Graphene-containing electrochemical biosensors can be used advantageously for the detection of the Sars-Cov-2 virus. The detection mechanism involves the immobi-lization of specific biomolecules on the surface of the graphene, capable of interacting with the virus, causing a change in the electrical properties of the graphene [47]. To be used in these biosensors, graphene must first be functionalized, creating functional groups on its surface, such as carboxylic acids, amines and thiols, etc., capable of re-acting and therefore immobilizing biomolecules, such as antibodies or nucleic acids, sensitive to the SARS- Cov-2. Graphene has shown itself to be very sensitive towards the detection of the virus thanks to its unique properties such as the high surface area and its high conductivity. The use of graphene in biosensors has proved to be more advantageous than other nanomaterials such as gold nanoparticles, silicon being cheaper, and offering greater sensitivity and therefore faster response times. Despite these evident advantages in the use of graphene for the realization of biosensors for the detection of the Sar-cov-2 virus, an intense research phase is still needed to opti-mize, for example, the functionalization phases of graphene and more generally a de-velopment of more convenient methods of graphene production so that these biosen-sors can be increasingly cheaper and widely commercialized [48,49].”

  • There is a lack of more perspective on the prospects of nanomaterials in fighting viruses. There needs to be more discussion about the issues that will arise in these areas.

4.R) In the manuscript the following part has been added (Lines 687-724):

“According to the European Chemicals Agency (ECHA), nanotechnologies will be increasingly present in daily life and this sector is expected to develop very rapidly which will make a significant contribution to industrial and economic growth in the coming years. According to widespread scientific opinion, nanotechnologies will go through 4 stages of development. The first two are already underway and extensively developed. The first concerns the use of nanomaterials in "passive nanostructures" which can be the creation of nanoreinforced nanomaterials or the application of nanocoatings that can be used for example for the creation of self-cleaning or antimicrobial surfaces. The second phase, which has already begun, is aimed above all at what could be defined as "active nanostructures", i.e. capable, thanks to their bioactivity, of being used as target drugs, for example by coating the nanoparticles with specific proteins capable of interacting with the virus, the bacterium, the organ or with the diseased cell. The third and fourth phases will be more complex and highly sophisticated and will concern the creation of nanosystems such as for nanorobotics to then move on to the generation of molecular nanosystems capable, for example, of using energy and promoting a series of specific functions such as intercepting and virus elimination. Current research has clearly shown the important role that nanomaterials, as well as nanoparticles, can play in the fight against viral diseases and especially against Covid-19. Nanotechnology paves the way for the research of new drugs and treatments against covid-19 by addressing the problems of low biocompatibility, poor stability and toxicity issues. Nanotechnology applications, however, present some critical issues that need to be addressed to facilitate their wider implementation in the healthcare field. It is important to underline that particular attention must be paid to the behavior of nanomaterials when they come into contact with the human body, for example inhaled or entering the bloodstream, where they can potentially change their behaviour. Most of the studies reported in the literature have evaluated biocompatibility only through laboratory and in vitro tests and using generic protocols. Precise standardized protocols for characterizing the properties of nanomaterials, with particular reference to their behavior with respect to human health, are desirable. Overcoming these challenges requires effective collaboration between experts in materials science, pharmacology, toxicology and regulatory agencies. Another obstacle to overcome is the production of nanomaterials on a large scale in order to make possible a feasible and economic commercialization of nanoparticle-based products.

In conclusion, it can be said that nanotechnologies have already amply highlighted the potential ability to improve various aspects that are crucial for the fight against Covid-19, such as diagnostics, therapies and protection.”

  • A table or TOC is highly recommended to summarize the main points in this review.

5.R) Summary Table 2 has been included in the manuscript

Reviewer 2 Report

From the title and abstract, it is expected that an in-depth discussion and analysis will be presented of nano particle use in the fight against viruses in the recent pandemic. However, unfortunately that is not available. The authors have largely presented superficial information, only mentioning the use of nano particles without explanation of principles, mechanisms, advantages vs. limitations, and if the technologies mentioned were ever available for consumer use on a commercial scale. In one paragraph of the conclusions, the authors raise the commonly expressed concerns regarding potential health impacts and toxicity concerns on nano particles, but do not mention any investigations of these impacts in the discussed technologies.

Examples of where greater depth of information on mechanism and analysis of performance would be of use include:

Their use in sensors (section 2)

Laser deposition, with respect to Figure 4

Graphene oxide insertion and bombardment with Fe3+ ions (line 273)

The above cited points are only examples and a complete revision of the article is suggested, as the subject will be of interest to many readers.

Author Response

Dear Reviewer,

thank you for your time spent reviewing our manuscript and for giving us important tips for improving it. The manuscript has been revised in many parts. The changes have been highlighted in yellow. Here are the answers to your suggestions point by point. We hope that in this form the manuscript will find your approval to be considered for its publication.

We thank you and send you our best regards.

The authors

Comments and Suggestions for Authors

Comments and Suggestions for Authors

*) From the title and abstract, it is expected that an in-depth discussion and analysis will be presented of nano particle use in the fight against viruses in the recent pandemic. However, unfortunately that is not available. The authors have largely presented superficial information, only mentioning the use of nano particles without explanation of principles, mechanisms, advantages vs. limitations, and if the technologies mentioned were ever available for consumer use on a commercial scale. In one paragraph of the conclusions, the authors raise the commonly expressed concerns regarding potential health impacts and toxicity concerns on nano particles, but do not mention any investigations of these impacts in the discussed technologies.

*Reply.) The following parts have been added to the text:

(Lines 638-686) “The various nanomaterials reported in this review first of all show that they are all capable of contributing in a positive and effective way to countering the spread of the Sars-Cov-2 virus, although their way of acting, their characteristics are often different.

Graphene, graphene oxide and nanofibers mainly act as real filters capable of blocking the virus, thanks to the size of the pores present in their structures which are smaller than the size of the Sars-Cov-2 virus. So in this respect it is more precise to talk about blocking the virus.

Metallic nanoparticles act differently, releasing ions that come into contact with the genetic material of the virus, thus leading to its inactivation. So in the case of metal nanoparticles it is more correct to speak of inactivation of the virus. Beyond this sub-stantial difference in the mechanism of action, there are differences even within these two categories of nanomaterials.

Graphene and polymeric nanofibres, in general, have a water-repellent behavior, and therefore exhibit an action of removing droplets, which, as is known, are small water droplets and are among the most common carriers of the virus. These materials therefore have a double action: filtering of the virus that manages to reach the surface and preventive action against droplets.

Graphene oxide, on the other hand, has a hydrophilic nature, therefore it has no action to contrast droplets but only a virus blocking action. In fact, in the creation of masks, its use is mainly intended for internal layers. Graphene has another important characteristic that of the photothermic effect, i.e. if subjected to sunlight it heats up leading to the death of the virus. Graphene therefore presents itself, under this aspect, as one of the best materials exhibiting virus blocking properties, preventive droplet contrast and virus inactivation after exposure to sunlight. This means that the graphene masks could potentially be reused after sterilization in sunlight. As far as nanoparticles are concerned, almost all of them have a high virus deac-tivation efficacy which can reach up to 99.9%. The data reported in the literature still do not allow for the creation of a precise scale of efficacy as the laboratory tests are not yet standardised. However, it is possible to say that the effectiveness of nanoparticles depends on their size, shape, nanoparticle/support ratio and the construction charac-teristics of the nanomaterial.

Other important features of the face masks against the Sars-Cov-2 virus, in addi-tion to effectiveness, are comfort, breathability and safety.

Safety is a fundamental aspect, but to date with respect to nanomaterials there is a scarcity of information against a high level of pervasiveness on the market. These gaps are to be attributed to a certain fragmentation of the studies available so far in the lit-erature, which do not find a real common line. A common fear is that the nanomaterials inserted into the masks could be released and come into contact with the human body [98,99]. In most cases, however, manufacturers ensure that the insertion of nanomaterials inside the masks has been carried out with technologies, often covered by patent se-crets, capable of preventing any release harmful to human health.

Comparing the materials presented, with respect to the comfort and breathability of the masks, an important position is occupied by nanofibres. In fact, they themselves form a nanofibre membrane that allows high breathability, flexibility and therefore high comfort for the wearer. For other materials, a support is always necessary, such as synthetic or natural fabric, in which to insert the nanomaterials which, depending on their nature, can interfere with the breathability and comfort of the mask”.

(Lines 687-720) “According to the European Chemicals Agency (ECHA), nanotechnologies will be increasingly present in daily life and this sector is expected to develop very rapidly which will make a significant contribution to industrial and economic growth in the coming years. According to widespread scientific opinion, nanotechnologies will go through 4 stages of development. The first two are already underway and extensively developed. The first concerns the use of nanomaterials in "passive nanostructures" which can be the creation of nanoreinforced nanomaterials or the application of nanocoatings that can be used for example for the creation of self-cleaning or antimicrobial surfaces.

The second phase, which has already begun, is aimed above all at what could be defined as "active nanostructures", i.e. capable, thanks to their bioactivity, of being used as target drugs, for example by coating the nanoparticles with specific proteins capable of interacting with the virus, the bacterium, the organ or with the diseased cell.

The third and fourth phases will be more complex and highly sophisticated and will concern the creation of nanosystems such as for nanorobotics to then move on to the generation of molecular nanosystems capable, for example, of using energy and promoting a series of specific functions such as intercepting and virus elimination.

Current research has clearly shown the important role that nanomaterials, as well as nanoparticles, can play in the fight against viral diseases and especially against Covid-19. Nanotechnology paves the way for the research of new drugs and treatments against covid-19 by addressing the problems of low biocompatibility, poor stability and toxicity issues.

Nanotechnology applications, however, present some critical issues that need to be addressed to facilitate their wider implementation in the healthcare field. It is important to underline that particular attention must be paid to the behavior of nanomaterials when they come into contact with the human body, for example inhaled or entering the bloodstream, where they can potentially change their behaviour. Most of the studies reported in the literature have evaluated biocompatibility only through laboratory and in vitro tests and using generic protocols. Precise standardized protocols for characterizing the properties of nanomaterials, with particular reference to their behavior with respect to human health, are desirable.

Overcoming these challenges requires effective collaboration between experts in materials science, pharmacology, toxicology and regulatory agencies.

Another obstacle to overcome is the production of nanomaterials on a large scale in order to make possible a feasible and economic commercialization of nanoparticle-based products.

In conclusion, it can be said that nanotechnologies have already amply highlighted the potential ability to improve various aspects that are crucial for the fight against Covid -19, such as diagnostics, therapies and protection”.

*) Examples of where greater depth of information on mechanism and analysis of performance would be of use include:

*)Their use in sensors (section 2)

*Replay) The following part has been added in the manuscript:

(Lines 128-150) “Graphene-containing electrochemical biosensors can be used advantageously for the detection of the Sars-Cov-2 virus. The detection mechanism involves the immobi-lization of specific biomolecules on the surface of the graphene, capable of interacting with the virus, causing a change in the electrical properties of the graphene [47]. To be used in these biosensors, graphene must first be functionalized, creating functional groups on its surface, such as carboxylic acids, amines and thiols, etc., capable of reacting and therefore immobilizing biomolecules, such as antibodies or nucleic acids, sensitive to the SARS- Cov-2 (Fig. 2). Graphene has shown itself to be very sensitive towards the detection of the virus thanks to its unique properties such as the high surface area and its high conductivity. The use of graphene in biosensors has proved to be more advantageous than other nanomaterials such as gold nanoparticles, silicon being cheaper, and offering greater sensitivity and therefore faster response times. Despite these evident advantages in the use of graphene for the realization of biosensors for the detection of the Sar-cov-2 virus, an intense research phase is still needed to opti-mize, for example, the functionalization phases of graphene and more generally a de-velopment of more convenient methods of graphene production so that these biosen-sors can be increasingly cheaper and widely commercialized [48,49].”

*) New figure2 have been added

**Laser deposition, with respect to Figure 4

**Replay) The following part has been added in the manuscript:

(Lines 261-274) “Specifically, the laser uses a high-energy and precision collimated light beam ca-pable of inducing specific morphological modifications on the surface of the target material. The laser coating allows the so-called graphene-like -layers (GL) to be depos-ited on the support of the mask, which consist of stacked layers of graphene-like na-nometric dimensions obtained from a controlled top-down demolition of a target graphite. Generally, to obtain an ultra-thin coating it is possible to use various tech-niques, many of which involve the use of solvents, which could damage or alter the surface to be coated. Laser technique offers advantageous aspects for the realization of coatings on biomedical materials such as facial masks, thanks to the possibility of con-trolling various process parameters that could damage the support on which the depo-sition takes place. In fact, it is not always easy for an implantable material to have all the required characteristics, for example it can have the right mechanical characteris-tics but be sensitive to solvents. The laser deposition system can preserve the function-ality of the material to be treated by not requiring the use of solvents in direct contact with the surface to be coated.”

**R.) Figure 3 has been improved

***) Graphene oxide insertion and bombardment with Fe3+ ions (line 273)

***Replay) The following part has been added in the manuscript:

(Lines 314-330) “Hu et al., reported the creation of a cotton fabric, with antimicrobial properties, on which graphene oxide was inserted and subsequently implanted with Fe3+ ions [75].

Ion implantation is a technique capable of modifying the properties, structure and morphology of surfaces of carbon-based materials using ions with different energies and doses in controlled way.

Specifically, a cotton fabric was treated with an aqueous dispersion of graphene oxide and placed in a microwave oven for programmed times. The fabric containing graphene oxide, obtained after drying, was subjected to ion implantation of Fe3+ using an ion implanter, with programmed ion doses and acceleration voltage.

The antimicrobial tests showed that the cotton fabric, containing graphene oxide and implanted with Fe3+ ions had a greater activity than the untreated ones and that increasing the dose of Fe3+ showed a greater antimicrobial activity. Although the tests have not been performed strictly on the SARS-Cov-2 virus, this study is important as regards the technique performed and the results of antimicrobial activity that the treated tissues have shown. Given the results obtained, it can be expected that these fabrics can also be efficient for the SARS-Cov-2 Virus in the same way and therefore it is to be hoped that there will be an in-depth analysis of the research activities on this study.”

Round 2

Reviewer 2 Report

Many thanks for addressing my concerns in the revision of your paper. I think the manuscript is acceptable in its present form and will be of great interest to readers of the journal.

The only request I have pertains to the following text, lines 679-681:
"In most cases, however, manufacturers ensure that the insertion of nanomaterials inside the masks has been carried out with technologies, often covered by patent secrets, capable of preventing any release harmful to human health."

Could you please comment if there have been independent tests of the manufacturer claims on product safety?

Author Response

Dear Reviewer,

 once again thank you for the advice and the time you have spent to make our manuscript better.

Following your last submitted suggestion:

*) The only request I have pertains to the following text, lines 679-681:

"In most cases, however, manufacturers ensure that the insertion of nanomaterials inside the masks has been carried out with technologies, often covered by patent secrets, capable of preventing any release harmful to human health." Could you please comment if there have been independent tests of the manufacturer claims on product safety.

We thought to add this sentence in the manuscript ((highlighted in yellow):

Replay:

(Lines 682-684) “Most of the products put on the market, according to the declarations of the producers, are tested and certified by independent bodies, while for the products in the research phase, the tests are carried out as a preliminary phase, in the laboratory”.

Best regards

The authors